# Discovery of GOT1 Inhibitors from a Marine-Derived *Aspergillus terreus* That Act against Pancreatic Ductal Adenocarcinoma

**DOI:** 10.3390/md19110588

**Published:** 2021-10-20

**Authors:** Shan Yan, Changxing Qi, Wei Song, Qianqian Xu, Lianghu Gu, Weiguang Sun, Yonghui Zhang

**Affiliations:** 1Hubei Key Laboratory of Natural Medicinal Chemistry and Resource Evaluation, School of Pharmacy, Tongji Medical College, Huazhong University of Science and Technology, Wuhan 430030, China; yanshanwhu@163.com (S.Y.); qichangxing@hust.edu.cn (C.Q.); Xuxuqq@hust.edu.cn (Q.X.); gulianghu@hust.edu.cn (L.G.); 2Guangdong Provincial Key Laboratory of Microbial Culture Collection and Application, State Key Laboratory of Applied Microbiology Southern China, Institute of Microbiology, Guangdong Academy of Sciences, Guangzhou 510070, China; 20191103@zcmu.edu.cn

**Keywords:** natural products, GOT1 enzyme, X-ray crystallography, pancreatic ductal adenocarcinoma cells, glutamine metabolism

## Abstract

Pancreatic ductal adenocarcinoma (PDAC) is a devastating digestive system carcinoma with high incidence and death rates. PDAC cells are dependent on the Gln metabolism, which can preferentially utilize glutamic oxaloacetate transaminase 1 (GOT1) to maintain the redox homeostasis of cancer cells. Therefore, small molecule inhibitors targeting GOT1 can be used as a new strategy for developing cancer therapies. In this study, 18 butyrolactone derivatives (**1**–**18**) were isolated from a marine-derived *Aspergillus terreus*, and asperteretone B (**5**), aspulvinone H (AH, **6**), and (+)-3′,3′-di-(dimethylallyl)-butyrolactone II (**12**) were discovered to possess significant GOT1-inhibitory activities in vitro, with IC_50_ values of (19.16 ± 0.15), (5.91 ± 0.04), and (26.38 ± 0.1) µM, respectively. Significantly, the molecular mechanism of the crystal structure of GOT1–AH was elucidated, wherein AH and the cofactor pyrido-aldehyde 5-phosphate competitively bound to the active sites of GOT1. More importantly, although the crystal structure of GOT1 has been discovered, the complex structure of GOT1 and its inhibitors has never been obtained, and the crystal structure of GOT1–AH is the first reported complex structure of GOT1/inhibitor. Further in vitro biological study indicated that AH could suppress glutamine metabolism, making PDAC cells sensitive to oxidative stress and inhibiting cell proliferation. More significantly, AH exhibited potent in vivo antitumor activity in an SW1990-cell-induced xenograft model. These findings suggest that AH could be considered as a promising lead molecule for the development of anti-PDAC agents.

## 1. Introduction

Pancreatic ductal adenocarcinoma (PDAC), a highly aggressive malignancy, is predicted to become the second leading cause of cancer-related deaths by 2030, despite the fact that 5-year survival rates have consistently remained below 10% throughout the past several decades [1]. Alteration of metabolism is a key indicator for cancer cells to meet their biosynthesis requirements [2,3]. Recently, it has been demonstrated that the demand for glutamine (Gln) in PDAC cells is excessive, promoting proliferation and tumor growth through a non-canonical Gln metabolic pathway regulated by KRAS [4,5,6]. In detail, the Gln-derived aspartate (Asp) is transported to the cytoplasm, where glutamic-oxaloacetic transaminase 1 (GOT1) catalyzes the transformation of *α*-KG and Asp to Glu and oxaloacetate (OAA), respectively. Subsequently, continuous catalytic reactions of malate dehydrogenase 1 (MDH1) and malic enzyme 1 (ME1) convert OAA to pyruvate and yield NADPH (Figure 1A) [7,8]. PDAC cells rely on these responses to increase the NADPH/NADP^+^ ratio in order to maintain the intracellular redox state [9,10]. As an essential enzyme in this pathway, the inhibition of GOT1 activity is growth-inhibitory in PDAC cells, but well tolerated by normal cells [11]. GOT1 metabolic pathways have also been shown to play a role in other malignant tumor cells, including small-cell lung cancer, breast adenocarcinoma, and glioblastoma multiforme [12,13]. Hence, GOT1 inhibitors represent a promising tool for studying the targeting of the metabolism of PDAC and other Gln-dependent cancers. In addition, although the crystal structure of GOT1 is known, the complex structure of GOT1 and its inhibitors has never been obtained [14]. Therefore, starting with higher activity GOT1 inhibitors, and combining the study of the crystal structure of GOT1 inhibitors with pharmacological evaluation, is an important approach to developing anti-PDAC drugs [15].

Butyrolactones, a class of highly functional lignan derivatives, are known as luciferase inhibitors [16]. In our previous study, we identified a potent butyrolactone-type GOT1 inhibitor from soil-sourced *Aspergillus terreus* collected from the bottom of the Yangzi River, which could suppress PADC cells’ growth by interfering with glutamine metabolism [10]. In our continuing study of GOT1-inhibitory metabolites, the chemical constituents of a marine-sediment-derived *A. terreus* obtained from the South China Sea were investigated systematically, and 18 butyrolactone derivatives (**1**–**18**) were isolated from this fungus (Appendix A). The GOT1-inhibitory activity screening of **1**–**18** suggested that (+)-asperteretone B (**5**), aspulvinone H (AH, **6**), and (+)-3′,3′-di-(dimethylallyl)-butyrolactone II (**12**) exhibit outstanding GOT1-inhibitory activities in vitro, with IC_50_ values of (19.16 ± 0.15), (5.91 ± 0.04), and (26.38 ± 0.1) µM, respectively. Further GOT1-inhibitory assays of AH were performed, and the high-resolution crystal structure of the complex was also obtained, which revealed that AH is a pyridoxal 5-phosphate (PLP) competitor that binds to the active site of GOT1. Simultaneously, we found that AH could inhibit the growth of PDAC cells both in vivo and in vitro, and induce the apoptosis of cancer cells by inhibiting Gln metabolism. As a selective GOT1 inhibitor, the complex crystal structure of GOT1–AH and the biological activity evaluation of AH will shed new light on the design and development of agents for the treatment of PDAC. In this paper, we report the isolation of **1**–**18**, and the co-crystal structure of GOT1–AH, as well as the anti-PDAC biological evaluation of AH.

## 2. Results and Discussion

### 2.1. Structural Elucidation

On the basis of the optical values of **1**–**18**, they were not racemate; corresponding to the spectroscopic characterization—including the values of chemical shifts and other physical constants from the reported papers [17,18,19,20,21,22,23]—compounds **1**–**18** were identified as terrusnolide B (**1**), terrusnolide D (**2**), terrusnolide C (**3**), (+)-asperteretone C (**4**), (+)-asperteretone B (**5**), aspulvinone H (**6**), butyrolactone III (**7**), versicolactone B (**8**), butyrolactone II (**9**), asperlide A (**10**), asperlide B (**11**), (+)-3′,3′-di-(dimethylallyl)-butyrolactone II (**12**), isobutyrolactone V (**13**), butyrolactone V (**14**), butyrolactone VI (**15**), butyrolactone IV (**16**), butyrolactone I (**17**), aspernolide E (**18**) (Figure 1).

### 2.2. The GOT1-Inhibitory Activity of **1**–**18**

Based on our previous study, all of the isolates **1**–**18** were evaluated for their GOT1-inhibitory activity. Among the evaluated isolates, compound **6** (AH) exhibited the best inhibitory effect of GOT1 activity, with an IC_50_ value of (6.91 ± 0.04) µM; compounds **5** and **12** also exhibited GOT1-inhibitory effects, with IC_50_ values of (19.16 ± 0.15) and (26.38 ± 0.12) µM, respectively, while the other isolates showed no obvious activity up to 40 µM (Figure 2A–C). Microscale thermophoresis (MST) experiments further confirmed that AH can bind GOT1, with a *K_d_* value of (2.14 ± 0.47) µM (Figure 2D). These findings suggest that AH could be used as a lead compound for the design and development of GOT1 inhibitors. Structure–activity relationship analysis showed that the butyrolactone-type natural products possessing the 5-benzylidene-3-phenylfuran-2(5*H*)-one core might bear considerably stronger GOT1-inhibitory activity, and that the isopentenyl side chains in butyrolactones might also increase the inhibition of GOT1 activity.

### 2.3. Co-Crystal Structure of GOT1 and AH

The crystal structure of GOT1 has been known for many years, but the co-crystal structure of GOT1/inhibitor has never been reported. This stimulated us to obtain the complex structure of GOT1 and AH, in an attempt to accurately elucidate the specific mechanism of action between GOT1 and GOT1 inhibitors. After many attempts, the co-crystal structure of GOT1–AH was finally obtained, and crystallization of GOT1 in the presence of AH afforded the opportunity to gain more detailed insight into the binding mode of compound AH within the GOT1 active site. The resolution of the complex structure was obtained at 2.6 Å (PDB ID 6LIG). The statistics of data collection and refinement are listed in Appendix A. The overall structure contains one large (residues 68–300) and two small (residues 17–67, 301–411) domains, with a core architecture consisting of an extended mixture of *α*-helices and *β*-sheets through the active cavity of the protease, and has been previously described in detail (Figure 3A) [16,24]. The large regions in the dimer are fundamentally identical, while two active sites show different substrate accessibility [25]. We also tried to obtain the GOT1 complex with compounds **5** and **12**, but failed to do so.

In contrast to WT-GOT1 (PDB ID 6DND), which contains a PLP as a cofactor in each monomer, the crystals soaked showed that chain A bound AH at the cleft in the interface of the subunits, whereas a PLP was situated in chain B (Figure 3B,C and Appendix A). As expected, AH adopts a compact conformation in the bulk solvent of GOT1, which fully occupies the active site. The surface electrostatic potential indicates that the active pockets are electropositive, with some hydrophilic patches for substrate recognition (Appendix A). Structural comparison using the PDBeFold and DALI servers revealed that there are few changes in chain A between GOT1–AH and WT-GOT1, with a Z-score of 24.1 and root-mean-square deviation (RMSD) of 0.32 Å over 410 residues. Overlapping these two structures, it was found that AH competitively binds to the PLP cofactor site of GOT1 (Appendix A). The hydroxyl-butyrolactone core of AH is located in the same position as the phenolic oxygen of PLP, forming a rich interaction with surrounding amino acids. Moreover, the main difference is that Tyr71 and Ser297 in the opposite monomer constitute the active cavity of AH, which may play a role in stabilizing AH.

Detailed analysis showed that AH is stacked in residues Gly39, Thr110, Trp141, Asp223, Tyr226, and Phe361 by hydrophobic interactions, while the isopentenyl group in AH at the entrance interacts with Ser297 and Tyr71 from the other subunit (Figure 4A). In addition, the side chains of Gly109, Ser258, Arg267, and Asn195 are hydrogen-bonded to O4, O32, and O26 of AH, respectively (Figure 4B). Interestingly, the C17 atom of AH is covalently linked to the N atom of Lys259 in the large domain via an isopentyl linkage, similar to the PLP in WT-GOT1, which is connected to form an internal Schiff base via the oxygen atom on the aldehyde group [26]. According to the chemical structure of AH, it may form a strong π-π stacking effect with GOT1. In chain B, the residual electron density clearly reveals the presence of PLP, which forms hydrogen bonds with Thr110, Tyr226, Ser258, and Arg267 (Appendix A). The rich hydrogen-bonding network in the active site contributes to stabilizing the spatial structure of proteins and the performance of their physiological functions [27,28]. When the hydrogen-bonding network formed by PLP is disturbed, the cofactors cannot be activated, resulting in the reduction or ablation of GOT1 activity.

### 2.4. AH Inhibited Activity in Cancer Cells

To explore the antitumor activity of AH, several human cancer cell lines including pancreas (AsPC-1, PANC-1, SW1990), breast (HCC1806, MM231, MM453), colorectal (HCT116), and ovarian cancer cells (ES2), along with a nonmalignant human pancreatic duct epithelial cell line (HDPE6C7), were selected to detect the inhibitory effect of AH on their growth. After treatment for 48 h, AH exhibited significant cytotoxicity effects on the SW1990, PANC-1, and AsPC-1 cell lines compared to the other cell lines, with an IC_50_ ranging from 6.32 to 10.47 µM (Figure 5). Importantly, AH exhibited minimal cytotoxicity effects on the nonmalignant HPDE6C7 cell line, with an IC_50_ value over 100 µM. These data show that AH possesses selective anti-proliferation activity against PDAC cells.

### 2.5. AH Modulates Metabolism and ROS Response

GOT1 is a key enzyme that is involved in the production of Gln-dependent NADPH in PDAC, and catalyzes the reversible conversion of Asp to OAA. To investigate the effects of AH on Gln metabolism, the associated metabolites including OAA and malate (Mal) were detected. As shown in Figure 6A, AH treatment reduced the content of OAA and Mal, while increasing Asp correspondingly. Consequently, the ratio of NADPH/NADP^+^ was remarkably decreased (Figure 6B). As excessive reactive oxygen species (ROS) can enhance cellular oxidative stress—resulting in damage to DNA, proteins, or lipids, causing apoptosis or necrosis [29]—we examined the physiological correlation between AH treatment and ROS response. 2,7-Dichlorofluorescein diacetate (DCFDA) was used to measure the intracellular ROS levels. In SW1990 cells, the concentration of ROS was positively correlated with the consumption of AH, indicating that cells inhibit the activity of GOT1 by upregulating ROS levels (Figure 6C,D). The above data indicate that AH treatment could block the production of Gln-dependent NADPH and participate in the redox homeostasis of PDAC cells.

### 2.6. AH Treatment Affected the Apoptosis, Cell Cycle, Proliferation, and Migration in SW1990 Cells

In addition to inhibiting cell proliferation, we also investigated the mechanism of growth inhibition induced by AH, via flow cytometry. The double staining of FITC–Annexin V and propidium iodide (PI) revealed that AH could induce apoptosis in SW1990 cells in a dose-dependent manner. After 48 h incubation at concentrations of 0 µM, 10 µM, 20 µM, and 40 µM, the induction rates were 4.6%, 9.5%, 28.3%, and 80.2%, respectively (Figure 7A). Simultaneously, the cell cycle distributions of SW1990 cells were examined in the presence of increasing doses of AH treatment. As shown in Figure 7B, AH increased the cell number at the S phase in a dose-dependent manner, and led to a corresponding decrease in the G0/G1 phase. Meanwhile, there was only a slight increase in the G2/M phase when compared with high-dose AH treatment. In detail, AH could increase the proportion of S-phase cells (25.1%, 32.5%, 38.6%, and 43.9%, respectively), with a decrease in the G0/G1 phase (68.2%, 59.4%, 48.6%, and 43.5%, respectively) (Figure 7B). Subsequently, the EDU staining analysis and wound-healing cell migration test also proved the anti-proliferative effect of AH in SW1990 cells (Figure 7C–E), in a dose- and time-dependent manner. All of these data demonstrate that the S-phase arrest may be partially attributable to the reduction in the viability of PDAC cells by AH.

### 2.7. In Vivo Antitumor Activity of AH

Next, the antitumor effect of AH in vivo was evaluated in a CB-17/SCID murine xenograft model. One week after subcutaneous injection of PDAC cells into their lower flanks, the mice were subjected to intraperitoneal injection of vehicle or AH (2.5 mg/kg/d and 5 mg/kg/d) for 14 days. As presented in Figure 8A–D, the tumor volume of the experimental group was significantly smaller than that of the control group, suggesting that AH had an inhibitory effect on the growth of SW1990 xenografts. Detection of tumor tissue metabolites found that OAA and Mal decreased, while Asp and Gln increased correspondingly. Therefore, the proportion of NADPH/NADP^+^ was accompanied by a decrease, consistent with the effect in vitro (Figure 8E,F). Moreover, there were no significant changes in the body weight or histomorphology of the mice, indicating that AH is an effective and safe antitumor agent.

## 3. Materials and Methods

### 3.1. Fungal Material

The marine sediment was collected in the Sanya Bay of the South China Sea, Sanya, China, in July 2019 (summer). The strain was collected from the marine sediment in August 2019 (summer). According to the morphology of this fungus, combined with the sequence analysis of the ITS region, the strain was identified as an *Aspergillus terreus* (GenBank accession number: OK465110). This *A. terreus* was preserved in Tongji Medical College, Huazhong University of Science and Technology.

### 3.2. General Experimental Procedures, Fermentation, Extraction, and Isolation

The high-resolution electrospray ionization mass spectra (HRESIMS) were recorded in positive-ion mode on a Thermo Fisher LC-LTQ-Orbitrap XL instrument. ECD data were measured with a JASCO-810 CD spectrometer instrument. Optical rotations UV data, and IR data were recorded on a PerkinElmer 341 instrument, a Varian Cary 50 instrument, and a Bruker Vertex 70 instrument with KBr pellets, respectively. Semi-preparative HPLC was conducted on a Dionex HPLC system equipped with an Ultimate 3000 pump (Thermo Fisher, Scientific, Waltham, MA, USA), an Ultimate 3000 autosampler injector, and an Ultimate 3000 diode array detector (DAD) controlled by Chromeleon software (version 6.80), using a reversed-phased C18 column (5 μm, 10 × 250 mm, Welch Ultimate XB-C18). One- and two-dimensional NMR data were recorded on a Bruker AM-400 instrument, with the reference of ^1^H and ^13^C NMR chemical shifts of the solvent peaks for methanol-d_4_ (δ_H_ 3.31 and δ_C_ 49.0) and CHCl_3_-d (δ_H_ 7.26 and δ_C_ 77.0). Silica gel 60 F_254_ was used for TLC (thin-layer chromatography) detection, and spots were visualized by spraying heated silica gel plates with 5% H_2_SO_4_ in EtOH. Column chromatography (CC) was carried out using silica gel (80–120, 100–200, and 200–300 mesh, Qingdao Marine Chemical, Inc., Qingdao, China), LiChroprep RP-C_18_ gel (40–63 μm, Merck, Darmstadt, Germany), and Sephadex LH-20 (GE Healthcare Bio-Sciences AB, Bjorkgatan, Sweden). Experimental stains were incubated in potato dextrose agar (PDA) medium at 28 °C for 7 days to prepare the seed cultures, which were then transferred into 1000 mL Erlenmeyer flasks, each containing 400 g of rice (total 100 kg). After cultivation for 28 days, the medium was extracted with 95% aqueous EtOH five times at room temperature. Afterwards, the solvent was removed under reduced pressure to yield a total residue, which was then suspended in water and partitioned repeatedly with EtOAc. The EtOAc extract (1.2 kg) was chromatographed on silica gel CC (80–120 mesh) using an increasing gradient of petroleum ether–ethyl acetate (100:0 to 0:100) to produce seven fractions (A–G).

Fraction D (112.3 g) was fractioned on silica gel CC (200–300 mesh) using an increasing gradient of petroleum ether–ethyl acetate (50:1 to 5:1) to produce eight subfractions (D1–D8). Subfraction D3 (12 g) was chromatographed on Sephadex LH-20 eluted with CH_2_Cl_2_–MeOH (1:1, *v/v*) to yield four fractions (D3.1–D3.4). Fraction D3.2 (820 mg) was purified using semi-preparative HPLC eluted with MeCN–H_2_O (63:37, *v/v*, 2.0 mL/min), producing compounds **1** (112 mg), **2** (23 mg), **3** (41 mg), and **8** (7 mg). Fraction D3.3 was purified using semi-preparative HPLC (MeCN–H_2_O, 57:43, *v/v*, 2.0 mL/min) to yield compounds **5** (11 mg), **7** (16 mg), and **9** (22 mg). Subfraction D4 (9 g) was chromatographed on Sephadex LH-20 eluted with CH_2_Cl_2_–MeOH (1:1, *v/v*) to yield three fractions (D4.1–D4.3). Compounds **6** (126 mg), **10** (5 mg), **11** (2 mg), and **17** (32 mg) were purified using semi-preparative HPLC (MeCN–H_2_O, 50:50, *v/v*, 2.0 mL/min) from fraction D4.1. Fraction D4.2 was purified using semi-preparative HPLC (MeOH–H_2_O, 70:30, *v/v*, 2.0 mL/min) to yield compounds **4** (9 mg), **12** (7 mg), and **13** (2 mg). Fraction E (32 g) was fractioned on silica gel CC (200–300 mesh) using an increasing gradient of petroleum ether–ethyl acetate (50:1 to 1:1) to produce four subfractions (E1–E4). Subfraction E3 (4 g) was chromatographed using an RP-C_18_ column with MeOH–H_2_O (from 30:70 to 90:10, *v/v*) to produce four subfractions (E3.1–E3.4). Compounds **14** (11 mg), **13** (5 mg), **15** (7 mg), and **18** (8 mg) were purified from fraction E3.3using semi-preparative HPLC (MeCN–H_2_O, 31:69, *v/v*, 2.0 mL/min).

### 3.3. Materials

Antibodies against GOT1 and *β*-actin were purchased commercially from Santa Cruz Biotechnology; goat anti-mouse IgG and goat anti-rabbit IgG antibodies were purchased from Cell Signaling Technology. Benzyloxycarbonyl-Val-Ala-Asp fluoromethylketone (Z-VAD-FMK) was obtained from Selleck; 20,70-dichlorodihydrofluorescein diacetate (DCFH-DA) was obtained from Invitrogen.

### 3.4. Molecular Cloning, Expression, and Purification

The expression construct pET26b-got1 (Invitrogen, Carlsbad, NJ, USA) was C-terminal His_6_-tagged and transformed into *E. coli* BL21 (DE3) cells at 30 °C for 6 h. The recombinant protein was initially passed over a HisTrap™ FF crude column (GE Healthcare, 5 mL, Pittsburgh, PA, USA). Peak elution fractions containing GOT1 were collected and concentrated using centrifuge tubes (Millipore, Billerica, MA, USA) with a 10 kDa cutoff, and then purified via anion-exchange chromatography with Resource Q (GE Healthcare, 5 mL, Pittsburgh, PA, USA). Subsequently, the sample was injected onto a Superdex200 Increase 10/300 GL Column (GE Healthcare, Pittsburgh, PA, USA) and dialyzed in 20 mM HEPES buffer (pH 7.5) and 200 mM NaCl.

### 3.5. Enzyme Inhibition Assays

The GOT1 enzymatic activity assay was carried out using a 96-well plate reader as previously reported [10]. The 100 μL reaction system contained 0.1 mg/mL recombinant GOT1 protein, 1 mM α -KG, 1 mM NADH, 4 mM Asp, and 1 U/mL malate dehydrogenase, and measured absorbance at a wavelength of 340 nm. The degree of absorbance decrease was in direct proportion to GOT1 activity. The inhibitory activity in vitro was measured by adding different concentrations of compounds. SigmaPlot with the PlotEnzyme Kinetics Module was used to analyze the enzyme activity.

### 3.6. Microscale Thermophoresis

The Monolith NTTM Protein Labeling Kit was used to test the binding capacity of proteins to AH, according to the manufacturer’s instructions. GOT1 was labeled and diluted in a buffer containing 20 mM HEPES (pH 7.5) and Tween-20 (0.5 (*v/v*)%), and then incubated with different concentration of AH for 10 min at 37 °C. Subsequently, the materials were put into standard capillary tubes and tested for thermophoresis using a Monolith NT.115 instrument (NanoTemper Technologies, München, Germany). The parameters were set to temperature 25 °C, LED power 100%, and laser power 40%, with a 30 s on-time. NTAnalysis software was used to calculate the *K_d_* values.

### 3.7. Protein Crystallization

Protein crystals of GOT1 were grown via the sitting-drop vapor diffusion method after 48 h at 18 °C, with 0.25 μL of protein and 0.25 μL of precipitant solution consisting of 0.1 M HEPES pH 7.5, 0.005 M nickel(II) chloride hexahydrate, 0.005 M cobalt(II) chloride hexahydrate, 0.005 M magnesium chloride hexahydrate, 0.005 M cadmium chloride hydrate, and 12% *w/v* polyethylene glycol 3350. Before crystallization, WT-GOT1 (16 mg/mL) was mixed with AH at a final concentration of 40 μM and incubated on ice for 2 h, and then centrifuged to remove the precipitate. Formed crystals were transferred in mother liquor supplemented with 25% glycerol and flash-cooled in liquid nitrogen. Diffraction data of GOT1 were collected using a beamline BL-19U at the Shanghai Synchrotron Radiation Facility in China, and processed using the HKL-2000 package [22]. The structure of the complex was determined via molecular replacement using the human GOT1 (PDB entry 6DND) with the Phaser crystallographic software in a stepwise manner [16,30]. Subsequently, PHENIX and COOT software were used to manually refine and adjust the structural models [26,31]. PyMOL software was used to create the figures.

### 3.8. Cell Culture and Viability Assay

Pancreas (SW1990, AsPC-1, PANC-1), colorectal (HCT116), breast (HCC1806, MM231, MM453), and ovarian cancer cells (ES2), along with a nonmalignant human pancreatic duct epithelial cell line (HDPE6C7), were obtained from ATCC, and were cultured in 1640 medium or Dulbecco’s modified Eagle medium (DMEM) supplemented with 10% (*v/v*) fetal bovine serum (FBS), 100 μg/mL streptomycin, and 100 U/mL penicillin. Cells were maintained in a humidified atmosphere of 5% CO_2_ at 37 °C. MTT (3-(4,5-dimethylthiazol-2-yl)-2,5-diphenyl tetrazolium bromide) assay was used to assess the cell viability. Briefly, cells were seeded in 96-well plates at a density of 5 × 10^3^ cells per well for 24 h. Then, the medium was removed, and the cells were treated with different concentrations (0.1–100 μM) of AH, with DMSO as a vehicle control. After incubation for 48 h, 100 μL of MTT solution (2 mg/mL) was added to each well, and the wells were incubated at 37 °C for another 4 h. The formed formazan crystals were dissolved in DMSO (200 μL/well) and shaken for 5 min. Then, the absorbance of the solution was measured at 490 nm with a microplate reader. IC_50_ values were interpolated from the resulting dose-dependent curves using GraphPad Prism 7.0. The reported IC_50_ values are the average of three independent experiments, each consisting of six replicates per concentration level (overall *n* = 18).

### 3.9. Cell Proliferation Assay

Cells were seeded in 96-well plates at a density of 2 × 10^3^ cells per well. To remove OAA, cells were inoculated in complete medium (2 mM Gln and 10 mM Glu), which was then replaced the same day with culture medium supplemented with 10% FBS. The cells were stained with 0.1% crystal violet after being fixed in 10% formalin. After extracting the dye with 10% acetic acid, relative proliferation was determined by measuring the OD at 595 nm.

### 3.10. Cell Cycle Assay

The phase distribution of DNA content was determined via propidium iodide (PI) staining. Cells were treated with varying doses of AH or vehicle for 48 h, and then the cell pellets were stained in the dark for 30 min with 60 μg/mL RNase A and 30 μg/mL PI. Finally, CellQuest software (Becton-Dickinson, Franklin Lakes, NJ, USA) was used to perform flow cytometric studies. Three separate experiments were used to determine the cell cycle.

### 3.11. Cell Apoptosis Assay

The Annexin V–FITC apoptosis kit was purchased from USA BD PharMingen™. In brief, different concentrations of AH were added to the wells for 48 h, and then the cells were harvested, before being resuspended in Annexin-binding buffer and incubated with Annexin V–FITC (10 mg/mL) and PI in the dark for 15 min. Flow cytometry was used to detect the stained cells and analyze the apoptosis immediately.

### 3.12. Wound-Healing Assay

SW1990 cells were seeded in 6-well plates for 24 h. Each well was scratched with a 10 μL pipette tip, and then the plate was washed with PBS to peel off the suspended cells, before adding different concentrations of AH (0–40 μM). Images from the same area of the wound were taken separately under a microscope from 0 to 48 h.

### 3.13. Metabolomics Analysis

SW1990 cells were cultivated to around 50% confluence in DMEM media with 10 mM glucose, 2 mM glutamine, and 5% FBS. A complete medium was prepared prior to two hours of metabolite collection. Quantification kits were used to determine the abundance of Asp (Abcam, #ab102512, Cambridge, UK), Gln (BioVision, #K55, San Francisco, CA, USA), malate (BioVision, #K637, San Francisco, CA, USA), and OAA (Abcam, #ab83428, Cambridge, UK). According to the manufacturer’s instructions, we collected some cells and homogenized them in the buffers provided. The supernatants were centrifuged, and then deproteinized using a 10 K spin column before being analyzed and compared to standard curves.

### 3.14. Xenograft Studies

Animal treatment and experiments were conducted in the Experimental Animal Center. Commercial 8-week-old CB-17/SCID mice (male) were purchased from Huafukang, China. After being resuspended using 0.1 mL of PBS, SW1990 cells (3 × 10^6^) were injected subcutaneously into the lower flanks of the mice. When the tumors attained a diameter of ~3 mm, the mice were randomly divided into three groups—2.5 mg/kg, 5 mg/kg, and control—and were subsequently injected intraperitoneally with normal saline or AH every day. A Vernier caliper was used to measure the size of the tumors. After two weeks of therapy, mice were euthanized under deep anesthesia (chloral hydrate, 10%, 10 μL/g), and tumor tissue metabolites were detected using the above method. All animal studies were carried out in accordance with the Huazhong University of Science and Technology’s guidelines.

### 3.15. Statistical Analysis

The collected data were presented as mean ± SD. The statistical analysis was performed using the GraphPad Prism 7.0 program. ANOVA (one-way analysis of variance) used SPSS version 13.0 software to evaluate the multiple-group comparisons, with values of *p* < 0.05 deemed statistically significant.

## 4. Conclusions

In this study, 18 butyrolactone derivatives (**1**–**18**) were obtained from the solid culture extract of a marine-derived *Aspergillus terreus*. Among those isolates, aspulvinone H (AH, **6**) was found to bear remarkable GOT1 inhibitory activity in vitro, with an IC_50_ value of (5.91 ± 0.04) µM. Importantly, the crystal structure of GOT1–AH suggests that AH and the cofactor pyrido-aldehyde 5-phosphate competitively bind to the active sites of GOT1, likely representing the molecular mechanism of action between AH and GOT1. More notably, the crystal structure of GOT1–AH is the first reported complex structure of GOT1/inhibitor. Furthermore, in vitro biological studies indicated that AH can suppress glutamine metabolism, making PDAC cells sensitive to oxidative stress, and inhibiting cell proliferation; this metabolite also exhibited potent in vivo antitumor activity in an SW1990-cell-induced xenograft model. In our previous paper, we identified aspulvinone O (AO) as a potent butyrolactone-type GOT1 inhibitor from soil-sourced *A. terreus* collected from the bottom of the Yangzi River. AH featured a similar chemical structure and activity to those of AO, and both compounds possessed outstanding GOT1-inhibitory activity, and could suppress PADC cells’ growth by interfering with glutamine metabolism, illustrating the significant role of 5-benzylidene-3-phenylfuran-2(5*H*)-one-containing butyrolactones in the development of natural GOT1 inhibitors. In conclusion, the discovery of AH not only provides a promising lead compound for the development of novel GOT1-inhibitory agents, but also represents an attractive potential molecule for anti-PDAC drug development.

## Figures and Tables

**Figure 1 marinedrugs-19-00588-f001:**
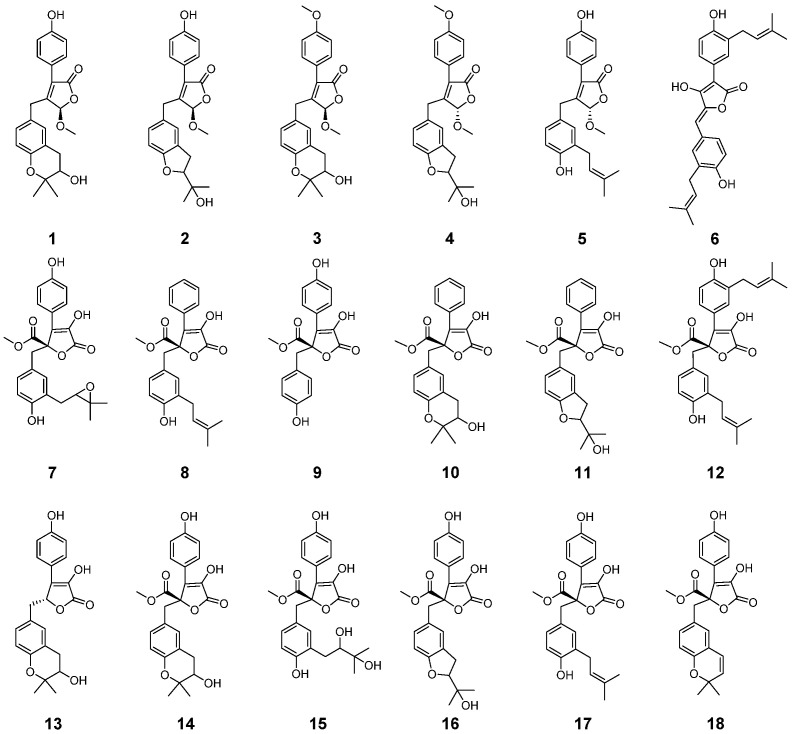
Structures of **1**–**18**.

**Figure 2 marinedrugs-19-00588-f002:**
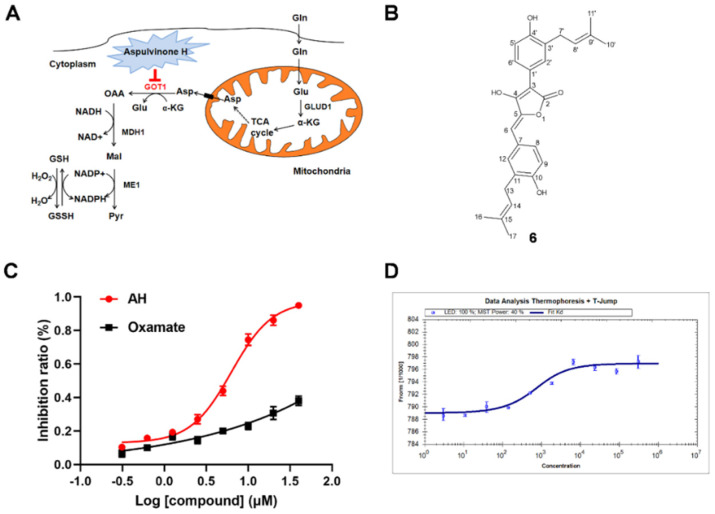
AH selectively targets GOT1: (**A**) Schematic overview of GOT1-mediated Gln metabolism. (**B**) The chemical structure of AH. (**C**) AH selectively inhibits GOT1 activity. (**D**) The *K_d_* value between AH and GOT1, as detected via MST assay.

**Figure 3 marinedrugs-19-00588-f003:**
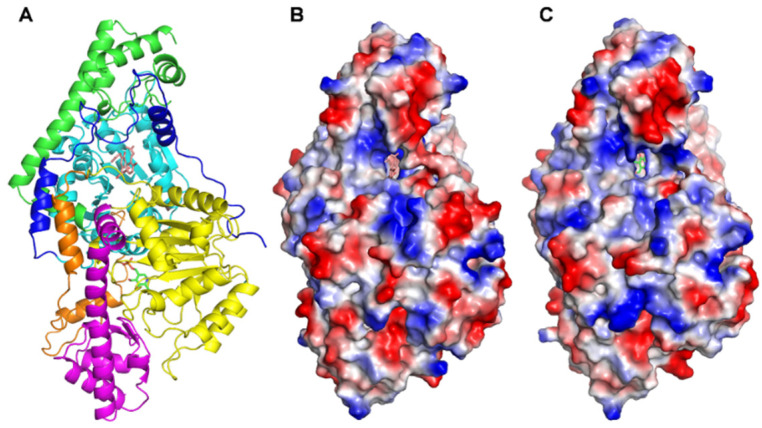
Structure of GOT1 in complex with AH: (**A**) Overall structure of GOT1-AH. The small domain 1 (blue/orange) includes two short α-helix structures; domain 2 (cyan/yellow) is a smaller (α/β)_7_ sandwich fold; domain 3 (green/magenta) is close to domain 1. Surface representation of (**B**) AH and (**C**) PLP in the active site of GOT1. Electrostatic potential shows that the active pocket is positively charged. Blue: positive potential; red: negative potential; white: neutral potential.

**Figure 4 marinedrugs-19-00588-f004:**
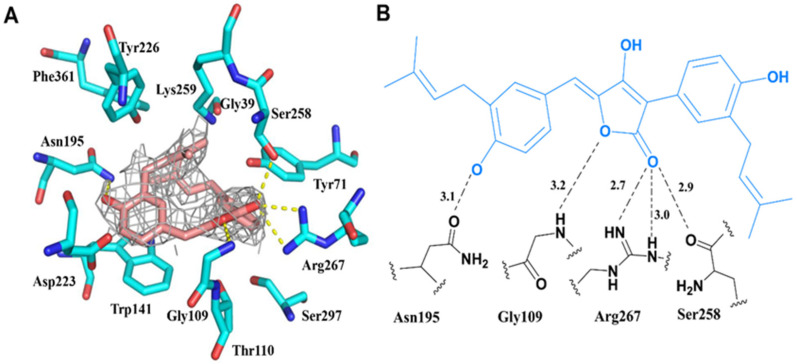
Detailed view of the active site: (**A**) The *2Fo-Fc* electron density of AH bound to chain A is shown as a gray mesh. The map is contoured at the 1σ level. Side chains involved in the binding site are shown. (**B**) Schematic of the potential hydrogen-bonding network between AH (blue) and GOT1 (black).

**Figure 5 marinedrugs-19-00588-f005:**
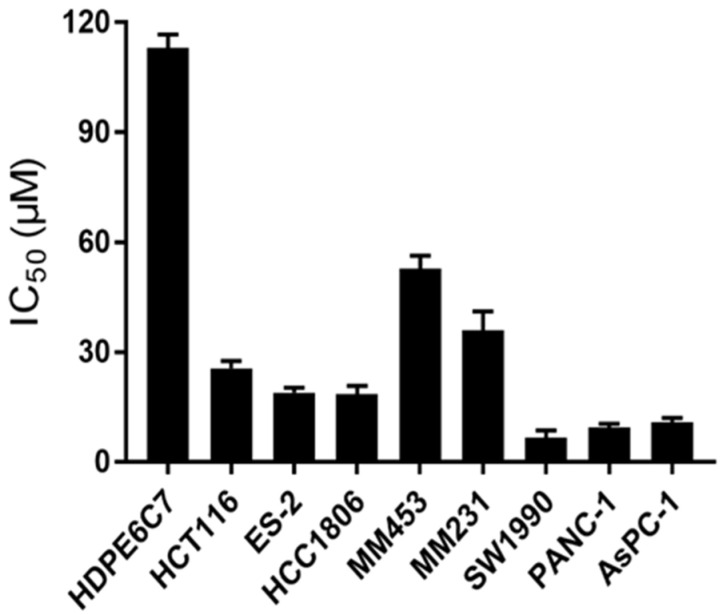
AH displays a higher growth inhibition rate in PDAC cells than in other cell lines. Pancreas cell lines (SW1990, AsPC-1, PANC-1), breast cell lines (HCC1806, MM231, MM453), HCT116, ES2, and a nonmalignant human pancreatic duct epithelial cell line (HDPE6C7) were treated with different concentrations of AH for 48 h, and the MTT assay was used to measure cell viability. IC50 values were interpolated from the resulting dose-dependent curves.

**Figure 6 marinedrugs-19-00588-f006:**
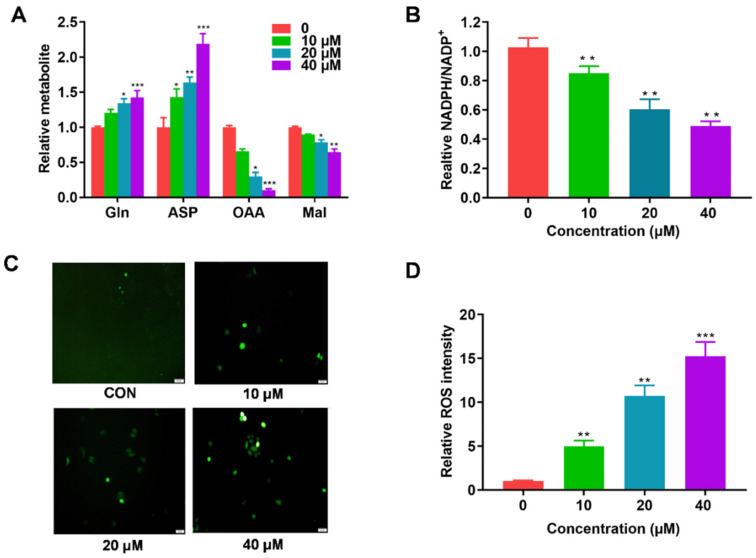
AH treatment induced Gln metabolism and maintained redox balance: (**A**) The effect of AH treatment (10, 20, or 40 µM) on relative metabolite abundance in SW1990 cells. (**B**) The effect of AH treatment (10, 20, or 40 µM) on the NADP^+^/NADPH ratio in SW1990 cells. (**C**,**D**) Intracellular ROS level measurement using carboxyH2DCFDA with 10, 20, or 40 µM of AH. Data represent means ± SD of three experiments. * *p* < 0.05, ** *p* < 0.01, *** *p* < 0.001 compared with the control group.

**Figure 7 marinedrugs-19-00588-f007:**
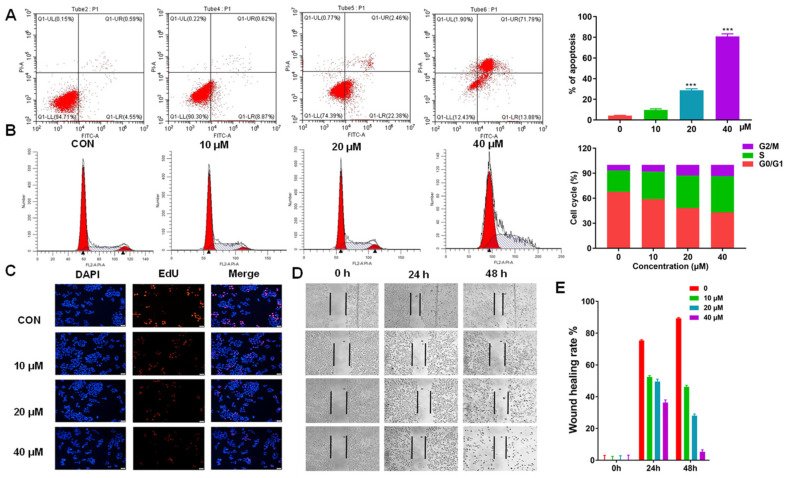
AH treatment affected the apoptosis, cell cycle, and proliferation in SW1990 cells: (**A**) Analysis of apoptosis in SW1990 cells using PI/Annexin V–FITC double staining after treatment with AH for 24 h. (**B**) Cell cycle distribution analysis of SW1990 cells upon treatment with AH. (**C**) DAPI and EDU double staining was used to assay the effect of AH on the proliferation of SW1990 cells. (**D**) Wound-healing assay in SW1990 cells with indicated AH treatments at 0 h, 24 h, and 48 h. (**E**) Quantitative analysis of wound-healing assay. Data represent means ± SD of three experiments. *** *p* < 0.001 compared with the control group.

**Figure 8 marinedrugs-19-00588-f008:**
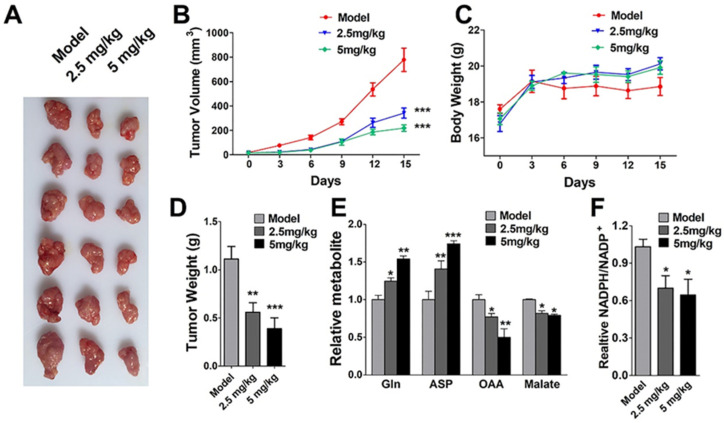
Evaluation of the in vivo efficacy of AH in xenograft tumors: (**A**) Tumor sample of xenograft mice after 14 days of treatment. (**B**) Tumor volume, (**C**) body weight, and (**D**) tumor weight for SW1990 xenografts in CB-17/SCID mice treated with 2.5 or 5 mg/kg/day of AH for 14 consecutive days. (**E**) Effects of metabolite abundance in different groups of tumor tissues. (**F**) NADPH/NADP+ ratio in different groups of tumor tissues. Analysis of variance (ANOVA) was used to compare the differences between groups. Data represent mean ± SD. * *p* < 0.05, ** *p* < 0.01, *** *p* < 0.001 compared with the control group.

## Data Availability

Not applicable.

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
