# Peer review of "Discovery of GOT1 Inhibitors from a Marine-Derived Aspergillus terreus That Act against Pancreatic Ductal Adenocarcinoma"

_marinedrugs, 2021, doi:10.3390/md19110588_

Round 1

Reviewer 1 Report

  1. Row#61- “marine mud-derived” please consider using a more precise phrase according to the exact sampling locations e.g. “marine sediment-derived” or “coastal mud-derived”
  2. Missing a key reference on asperteretones compounds that can be incorporated into row #78 to support the structure assignments.

DOI: 10.3389/fchem.2018.00422

  1. The authors employed solely semi-preparative HPLC to obtain what appear to be the R enantiomer of asperteretone B (compound 5) (Figure S1. Structures of 1-18). Do you isolate and assay the racemate of 5 or the purified R enantiomer of 5? The same query is also applied to compound 12.

Did you perform enantioselective HPLC purification of the isolated compounds prior to in vitro experiment? I do not find the chiral HPLC procedure mentioned in either MS or SI files.

  1. Please justify why did you prepare only the GOT1-AH complex? But not the complex with compounds 5 and 12.
  2. Row#88-13 typo??? Did you mean 12?

Author Response

List of responses

Responses to Reviewer 1:

Thanks for your valuable suggestions on corrections and modifications that addressed to the manuscript (ID: marinedrugs-1416094). In order to reach the requirement of Marine Drugs, we have tried our best to revise the whole manuscript, and the revised parts have been highlighted in yellow. Moreover, we have revised our manuscript according to your and all reviewers' suggestions. The detailed revisions were attached as following.

Comment 1: Row#61-“marine mud-derived” please consider using a more precise phrase according to the exact sampling locations e.g. “marine sediment-derived” or “coastal mud-derived”

Response: Thank you for your professional suggestion. We have revised the “marine mud-derived” to “marine sediment-derived”.

Comment 2: Missing a key reference on asperteretones compounds that can be incorporated into row #78 to support the structure assignments. DOI: 10.3389/fchem.2018.00422.

Response: We apologized for the missing of the key reference on asperteretones compounds. We have added the key reference as ref. 23 in the revised manuscript-R1. Thank you!

Comment 3: The authors employed solely semi-preparative HPLC to obtain what appear to be the R enantiomer of asperteretone B (compound 5) (Figure S1. Structures of 1-18). Do you isolate and assay the racemate of 5 or the purified R enantiomer of 5? The same query is also applied to compound 12.

Response: Thank you for your professional suggestion. We have not isolated and assayed the racemate of 5, and also not isolated and assayed the racemate of 12. Like Xuemei Niu et al only obtained the R orientated butyrolactone I, 3-sulfate- butyrolactone I and 4''-sulfate-butyrolactone I by semi-preparative HPLC (DOI: 10.1021/np070341r), Abdelaaty Hamed et al also only obtained the R orientated butyrolactones I-III (10.1080/14786419.2018.1544977), but Mengting Liu et al have reported the R/S orientated asperteretones A-D (10.3389/fchem.2018.00422).

Comment 4: Did you perform enantioselective HPLC purification of the isolated compounds prior to in vitro experiment? I do not find the chiral HPLC procedure mentioned in either MS or SI files.

Response: Thank you for your nice suggestion. Once we have isolated the compounds 1-18, we have tested their NMR data and optical values, and based on the former reported papers, we could fully confirmed those compounds were known compounds and the corresponding optical values identified they were not racemate. We have added the description in the revised manuscript-R1 guided by the nice reviewer 1.

Comment 5: Please justify why did you prepare only the GOT1-AH complex? But not the complex with compounds 5 and 12.

Response: Thank you! The culture of complex was so difficult and we have also try to obtain the GOT1 complex with compounds 5 and 12 but end in fail. AH possessing the best GOT1 inhibitory activity and was the most representative compound, so we only reported the GOT1-AH complex. As the careful reviewer 1 mentioned, we have added the description of the complex with compounds 5 and 12 in the manuscript-R1.

Comment 6: Row#88-13 typo??? Did you mean 12?

Response: We are so sorry for the mistake, this is a typo, we have corrected this mistake in the manuscript-R1. Thank you for your careful comment.

Reviewer 2 Report

  1. The manuscript was checked for text similarity and the plagiarism is too high in the manuscript. A copy f the text similarity report was attached to this review.
  2. The paper is well written and is accepted for publication after addressing these points:
  3. 1. A copy of the 1H and 13C NMR spectra of the compounds in the manuscript should be included in the supporting information.
  4. HPLC profile or purity report of all compound should be included in the supporting information of in the text.
  5. The citation of the reference in the text did not match Marine Drugs style 

Author Response

List of responses

Responses to Reviewer 2:

Thanks for your valuable suggestions on corrections and modifications that addressed to the manuscript (ID: marinedrugs-1416094). In order to reach the requirement of Marine Drugs, we have tried our best to revise the whole manuscript, and the revised parts have been highlighted in yellow. Moreover, we have revised our manuscript according to your and all reviewers' suggestions. The detailed revisions were attached as following.

Comment 1: The manuscript was checked for text similarity and the plagiarism is too high in the manuscript. A copy f the text similarity report was attached to this review.

Response: Thank you for your professional suggestion. We have tried our best to decrease the similarity of our manuscript.

Comment 2: The paper is well written and is accepted for publication after addressing these points: A copy of the 1H and 13C NMR spectra of the compounds in the manuscript should be included in the supporting information.

Response: We apologized for the missing of the 1H and 13C NMR spectra of the compounds. We have added the data in the revised supporting information-R1. Thank you so much!

Comment 3: HPLC profile or purity report of all compound should be included in the supporting information of in the text.

Response: Thank you for your kind suggestion. We have added the HPLC profile of all compound in the revised supporting information-R1 to confirm the purity of the isolates.

Comment 4: The citation of the reference in the text did not match Marine Drugs style.

Response: Thank you, we have checked and revised the wrong style citation of the reference in the revised manuscript-R1.

Reviewer 3 Report

 Shan Yan, et al reported the isolation of 18 butyrolactones from a marine-derived Aspergillus terreus, and that asperteretone B, aspulvinone H (AH) and (+)-3',3'-di-(dimethylallyl)-butyrolactone II have been discovered to possess significant GOT1 19 inhibitory activities in vitro. I have the following comments to be addressed before moving forward:

Experimental/supplementary

On what basis did you select fraction D to work on?

First, you need to add the HPLC profile for the extract that you got the 18 Butyrolactones from in the SI. At least I need to see the D3, D4, E3 subfractions.

Second, you need to add the 1H NMR data for all the 18 isolated Butyrolactones in the SI

Section 3.1. The fungus needs proper identification, please provide GenBank accession number and phylogenetic analysis/tree for your ITS sequence.

Author Response

List of responses

Responses to Reviewer 3:

Thanks for your valuable suggestions on corrections and modifications that addressed to the manuscript (ID: marinedrugs-1416094). In order to reach the requirement of Marine Drugs, we have tried our best to revise the whole manuscript, and the revised parts have been highlighted in yellow. Moreover, we have revised our manuscript according to your and all reviewers' suggestions. The detailed revisions were attached as following.

General Comment: Shan Yan, et al reported the isolation of 18 butyrolactones from a marine-derived Aspergillus terreus, and that asperteretone B, aspulvinone H (AH) and (+)-3',3'-di-(dimethylallyl)-butyrolactone II have been discovered to possess significant GOT1 19 inhibitory activities in vitro. I have the following comments to be addressed before moving forward.

Response: Thank you for your kind comments we have revised our manuscript according to your nice suggestions.

Comment 1: On what basis did you select fraction D to work on?

Response: Thank you so much for you nice comment! We have also isolated the fractions B, C and F, but we have not obtained the pure compounds, and from fractions D and E, we have isolated the compounds 1-18.

Comment 2: First, you need to add the HPLC profile for the extract that you got the 18 Butyrolactones from in the SI. At least I need to see the D3, D4, E3 subfractions.

Response: We have added the HPLC profile for the extract in the SI based on your nice comment.

Comment 3: you need to add the 1H NMR data for all the 18 isolated Butyrolactones in the SI.

Response: We apologized for the missing of the 1H and 13C NMR spectra of the compounds. We have added the data in the revised supporting information-R1. Thank you so much!

Comment 4: Section 3.1. The fungus needs proper identification, please provide GenBank accession number and phylogenetic analysis/tree for your ITS sequence.

Response: Thank you, we have added the GenBank accession number and ITS sequence in the revised manuscript-R1 and SI.

Round 2

Reviewer 1 Report

Please show optical rotation values for compounds 1-8 in the SI file

Author Response

List of responses

 Responses to Reviewer 1:

Thanks for your valuable suggestions on corrections and modifications that addressed to the manuscript (ID: marinedrugs-1416094). In order to reach the requirement of Marine Drugs, we have tried our best to revise the whole Supplementary Material. The detailed revisions were attached as following.

Comment 1: Please show optical rotation values for compounds 1-18 in the SI file.

Response: Thank you for your professional suggestion. We have added the optical rotation values for compounds 1-18 in the revised Supplementary Material-R2.

Reviewer 3 Report

Thanks for addressing my comments

Author Response

Thank you for your nice comments!